# AVCaps: An Audio-visual Dataset with Modality-specific Captions

## Abstract

In this paper, we introduce AVCaps, an audio-visual captioning dataset that contains separate textual captions for the audio, visual, and audio-visual contents of video clips. The dataset contains 2061 video clips constituting a total of 28.8 hours. We provide up to 5 captions for the audio, visual, and audio-visual content of each clip, crowdsourced separately. Existing datasets focus on a single modality or do not provide modality-specific captions, limiting the study of how each modality contributes to overall comprehension in multimodal settings. Our dataset addresses this critical gap in multimodal research by offering a resource for studying how audio and visual content are captioned individually, as well as how audio-visual content is captioned in relation to these individual modalities. To counter the bias observed in crowdsourced audio-visual captions, which often emphasize visual over audio content, we generated three audio-visual captions for each clip using our crowdsourced captions by leveraging existing large language models (LLMs). We present multimodal and crossmodal captioning and retrieval experiments to illustrate the effectiveness of modality-specific captions in evaluating model performance. Notably, we show that a model trained on LLM-generated audio-visual captions captures audio information more effectively, achieving 14% higher Sentence-BERT similarity on ground truth audio captions compared to a model trained on crowdsourced audio-visual captions. We also discuss the possibilities in multimodal representation learning, question answering, developing new video captioning metrics, and generative AI that this dataset unlocks. The dataset will be freely available online.

## 1 Introduction

The growth of multimodal large language models (LLMs) and multimedia content has created a growing demand for datasets and methods capable of a comprehensive understanding of audio-visual data Nagrani et al. (2022). In this context, captioning datasets play an important role in advancing AI and machine perception research helping models learn to interpret and describe complex audio-visual scenes. These datasets enable machines to generate meaningful descriptions, bridging the gap between humans and machines, and are essential for advancing tasks like image or video captioning Xu et al. (2015), multimodal scene understanding Li et al. (2022), and improving human-computer interaction Baltrušaitis et al. (2019). Specifically, audio-visual datasets with accompanying captions enable models to learn and interpret the rich, multimodal information in such data.

Multimodal perception models developed using captioning datasets have diverse real-world applications. For example, audio-visual captioning enhances accessibility by providing transcriptions and descriptions for multimedia content, making it accessible to individuals with disabilities Mocanu et al. (2019). They are also essential in automating subtitle generation Chen et al. (2023), improving content searchability Dong et al. (2018), supporting educational tools Zhou et al. (2018), and video analysis in security and surveillance Yuan et al. (2024).

Captioning datasets typically focus on descriptions for specific input modalities, such as audio, image, or video. Image captioning datasets like MS COCO Chen et al. (2015), and Flickr30k Young et al. (2014) provide textual descriptions for images that include various objects and people engaged in everyday activities. These datasets enable the development of models that can generate textual descriptions for static visual content.

Similarly, audio captioning datasets, such as AudioCaps Kim et al. (2019), Clotho Drossos et al. (2020), and MACS Martın-Morató & Mesaros (2021) provide textual descriptions that capture the acoustic scenes in audio clips. AudioCaps has captions for audio files from AudioSet Gemmeke et al. (2017), a dataset for a general-purpose audio classification from YouTube. Clotho contains captions for sounds occurring in nature such as water, birds, rain, etc., sourced from FreeSound Font et al. (2013) while avoiding sounds such as music, speech, and sound effects. MACS provides captions for audio files recorded in different European cities, from three acoustic environments - airports, public squares, and parks.

While these datasets have significantly advanced the field of multimedia understanding, they concentrate on describing one input modality, making it challenging to model the relationships between different modalities to get a complete context. Image captions emphasize objects present in the images describing spatial relationships and object properties. Audio captions contain descriptions of sound sources, environments, and temporal patterns present in the audio. For instance, consider a video clip of a dog barking at the door while a bell rings in the background. An image captioning dataset might focus solely on the visual aspect, describing the scene as "A dog is standing near a door", without acknowledging the audio elements. On the other hand, an audio captioning dataset might describe the sound as "A dog barking and a bell ringing" without any context of the visual scene. The lack of integration between the two modalities means that the complete context - "The dog is barking at the door because someone rang the bell" is not captured. This demonstrates the need for datasets that can handle both modalities simultaneously to generate richer and more contextually accurate captions.

Video captioning datasets, like MSR-VTT Xu et al. (2016) and ActivityNet Captions Krishna et al. (2017) offer captions that describe the content of video clips. MSR-VTT consists of 10000 video clips from various categories such as people, animation, TV shows, vehicles, etc., and 20 associated crowdsourced English captions per clip. ActivityNet Captions consists of video clips capturing various human activities annotated with natural language captions. Although these datasets include audio content as a part of the video clips, the primary focus of the descriptions is on the visual content. Winter (2019) also presents a study of how the descriptions vary depending on the sensory input and demonstrates visual as the dominant sensory input. Hence, current captioning datasets hamper our ability to understand how multimodal models perceive and integrate information from different modalities, and to quantify the extent to which each modality contributes to the overall representation.

To address these challenges, there is a pressing need for a dataset that provides captions for audio, visual, and audio-visual content for the same video clips. To this end, we designed a new audio-visual captioning dataset AVCaps, that contains modality-specific captions for audio, visual, and audio-visual content of the same video clips.

In this paper, our contributions are threefold:

- We introduce a novel audio-visual captioning dataset AVCaps, that contains 2061 videos, spanning a total duration of 28.8 hours, and includes crowdsourced captions for audio, visual, and audio-visual content of the same video clips.

- Since human annotators tend to focus more on visual elements than audio when describing audio-visual content Winter (2019), we also present as part of the dataset, three audio-visual descriptions for each clip generated by an LLM using the modality-specific crowdsourced captions.

- We develop audio-visual captioning and retrieval models on the AVCaps dataset and demonstrate how modality-specific captions can be used to enhance and evaluate the ability of these models to capture information from audio and visual modalities.

## 2 DATASET

In this section, we discuss the overall process involved in creating our dataset. This includes selecting videos, collecting human-annotated captions, data cleaning, and creating the data splits. We also present some unique characteristics of the collected captions that give a very good insight into how humans perceive different modalities.

## 2.1 DATA COLLECTION

The first step involved selecting videos for the dataset. Our goal was to choose publicly available videos that can be reused, redistributed, and encompass a broad range of topics. To achieve this, we sourced the videos from the VidOR dataset Shang et al. (2019); Thomee et al. (2016). The VidOR dataset comprises 10000 Creative Commons-licensed video clips from Flickr, featuring a variety of scenarios such as indoor and outdoor family moments, musical performances, speeches, pets, and babies. The VidOR dataset also contains annotations of 80 object classes with their spatiotemporal bounding boxes and 50 categories of action and spatial predicates that describe the relationships between different objects within the videos.

We filtered this dataset to select videos with at least three object classes and a minimum duration of 10 seconds to aid diverse and descriptive captions. After this filtering, we had 2176 videos for further data annotation. We isolated audio and visual content from the selected videos, to provide them as separate modalities for the annotators. Using Amazon Mechanical Turk (AMT), we crowd-sourced five captions per clip for the audio, visual, and audio-visual content in the selected clips, annotated by different workers. To ensure quality, we restricted the task to AMT workers with a 97% approval rate. Since we collected English captions, we limited the task to workers from English-speaking countries such as the USA, UK, and Australia. The workers were instructed to complete a Human Intelligence Task (HIT) by describing the input clip (audio, visual or audio-visual) using a complete sentence with at least five words in the caption. They were also provided with some example captions to assist them. The workers were compensated at a rate of $0.25 per HIT. In total, 4421 workers annotated the entire dataset. The instructions provided to the workers for annotating different modalities are provided in Appendix B.

## 2.2 DATA CLEANING

We implemented a two-stage data cleaning process to ensure high data quality.

**Stage 1: Automated Error Correction**

Following the crowdsourcing phase, each video clip was paired with 15 captions, 5 each for the audio, visual, and audio-visual content. Despite limiting the tasks to workers from English-speaking countries, a notable proportion of captions contained spelling and grammatical errors. To address these issues, we implemented an automated error correction step using the GPT-3.5 language model. Following the grammar correction prompt outlined in the OpenAI API documentation[1], the following template was employed:

```
You will be provided with statements.  Your task is to
convert them to standard English.

crowdsourced caption 1
crowdsourced caption 2
.
.
crowdsourced caption n
```

After this error correction process, we observed a decrease in the vocabulary of the corrected captions due to the removal of misspellings. The average length of the corrected captions increased, due to adding articles and determinants to the sentences. Additionally, readability indices Thomas et al. (1975) showed that the comprehension level required increased, from 7 years old to 9-10 years. Examples illustrating this error-correction process and its effects are provided in Appendix C.

**Stage 2: Manual Relevance Screening**

Next, we found that some captions were not relevant to the scenes, as the annotations were noisy. For instance, one worker repeated the caption "I enjoy it greatly" 10 times and another worker wrote "The voice is very nice and likable", 19 times. Some workers copied the example captions provided for reference. The audio modality captions were most affected with a total of 427 HITs completed using one of the example captions.

---

[1]https://platform.openai.com/docs/examples/default-grammar

In this cleaning stage, we manually read all the crowdsourced captions and checked if they were relevant to the corresponding input file, and removed noisy captions from the dataset that were not related to the input. We removed video clips and their corresponding audio and visual clips from the dataset that did not have at least one relevant audio, visual, and audio-visual caption.

After the data cleaning process, we ended up with 2061 clips, with a total duration of 28.8 hours. Each video varies in length, with the shortest being 20 seconds and the longest 175 seconds. The average duration of the videos is 50.45 seconds, while the median duration is 44.54 seconds. The total number of captions is 6.1k, 7.6k, and 7.7k for audio, visual, and audio-visual clips respectively. Some sample captions from the dataset are shown in Appendix A.

## 2.3 TRAINING, VALIDATION AND TEST SPLITS

The dataset is split randomly approximately into 80%-10%-10% for training, validation, and testing, respectively. The training set contains 1661 clips, while the validation and test set each consist of 200 clips. The data is split in such a way that all files in the validation and test splits have at least three ground truth captions for audio, visual, and audio-visual content of the clip while all the files in the training split have at least one caption for each category. The content of the videos in the training split contains 73 unique VidOR object categories while the validation and test videos have 49 and 48 respectively.

## 2.4 DATASET CHARACTERISTICS

The audio, visual, and audio-visual captions are analyzed separately and their statistics are presented in Table 1. The vocabulary size has been calculated by lemmatizing the words using the python NLP toolkit[2]. The most common sentence in the dataset is "The baby is playing a game", repeating 25, 12, and 11 times in audio, visual, and audio-visual captions respectively. The words "baby" and "play" are among the top three words in the audio, visual and audio-visual captions.

Table 1: Statistics of the crowdsourced audio, visual, and audio-visual captions. The average length is the average number of words in the caption. Parts of speech (POS) are also shown with percentage of nouns, verbs, adj. (adjectives), adv. (adverbs).

| Caption type | Total captions | Unique captions | Vocab. size | Average length | POS | | | |
|---|---|---|---|---|---|---|---|---|
| | | | | | Nouns | Verbs | Adj. | Adv. |
| Audio | 6110 | 5737 | 1343 | 10.27 | 56.7% | 35.0% | 4.6% | 3.2% |
| Visual | 7599 | 7253 | 1863 | 10.24 | 61.2% | 31.2% | 5.9% | 1.2% |
| Audio-visual | 7747 | 7487 | 1977 | 10.77 | 60.9% | 31.7% | 5.7% | 1.3% |

Parts of speech (POS) analysis gives us information about how the captions are structured. Captions are mainly formed using nouns and verbs, with a small proportion of adjectives and adverbs. In particular, audio captions exhibit higher frequency of verbs, indicating that audio is often associated with sound-producing actions Giordano et al. (2022). The workers use different sensory inputs to annotate a clip, resulting in different descriptions for the audio, visual, and audio-visual content of the clips. An example of how different annotators describe different sensory inputs is shown in Table 2. In audio-visual and visual captions, the actions described are related to visual sensory e.g."seeing" or "looking" and for the audio captions, they are related to acoustic sensory inputs, e.g. "talking" and "barking".

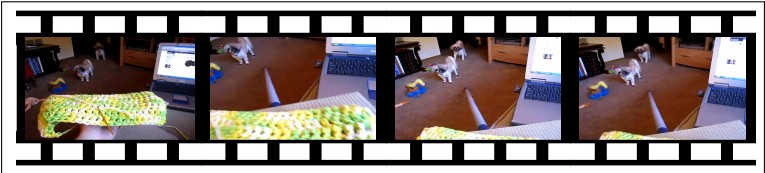

Figure 1: An example video from the dataset.

---

[2]https://www.nltk.org/

Table 2: Captions annotated for video clip shown in Figure 1.

| Input | Caption |
|---|---|
| Audio | A woman is sharing a humorous anecdote about her dog while the dog is barking loudly in the background. This lady is laughing out loud. |
| Visual | The two dogs are scared when things move. Two dogs are looking at a basket. Two dogs were listening and looking at the mat. |
| Audio-visual | There are two dogs that are seeing something. The dogs are playing with each other. The homeowner interacts with her puppies by showing them something and laughing. |

Some of the most frequently used adjectives in audio captions are: "unintelligible", "husky", "unclear" or "louder", which are related to describing acoustic content. Some examples of the most frequently used adjectives in the visual and audio-visual captions are: "ugly", "illuminated", "coloured", "oversized", and "attractive". The audio-visual captions contain more information about the visual content than the audio content, as it can be seen in the example shown in Table 2. The audio-visual captions do not focus on the *woman talking* or the *dog barking*.

## 2.5 LLM-GENERATED AUDIO-VISUAL CAPTIONS

As stated in Winter (2019), we also observed during the manual screening process that in many cases, the crowdsourced audio-visual captions, without explicit instructions to focus on a specific modality, tended to emphasize visual cues more than auditory ones. This observation further substantiated in Section 2.6, has motivated the generation of a new set of audio-visual captions that provide a balanced representation of the visual and audio content. To do so the crowdsourced audio, visual, and audio-visual captions were fed to the LLM (GPT-4o), and the model was prompted to give three audio-visual captions. The prompt given to the LLM for this task is as follows:

```
You will be provided with a few descriptions of audio, visual,
and audio-visual contents of a video clip.  These were described
by different people.  Hence, they may focus on different aspects
of the video clip when they are looking at the audio and visual
parts.  There could be some inconsistencies.  For example,
some people may refer a man in the visual clip as father, dad,
grandfather, person and so on.

Your task is to create three consolidated captions that capture
consistent information from audio, visual, and audio-visual
descriptions while ignoring some information that may have been
misinterpreted.

Audio captions:
crowdsourced caption 1
...
crowdsourced caption n

Visual captions:
crowdsourced caption 1
...
crowdsourced caption n

Audio-visual captions:
crowdsourced caption 1
...
crowdsourced caption n
```

The LLM-generated captions have on average 15.44 words per caption, almost 50% higher than the average number of words in the crowdsourced captions. Examples of audio-visual captions produced by GPT-4o based on ground truth captions are provided in Appendix 5.

## 2.6 COMPARISON OF GROUND TRUTH AND LLM-GENERATED AUDIO-VISUAL CAPTIONS

To support our claim that the crowdsourced audio-visual captions emphasize visual cues more than auditory ones, we analyzed the audio and visual information present by comparing them against the ground truth audio and ground truth visual captions. Similarly, we compared the LLM-generated audio-visual captions to the ground truth audio and visual captions. For this comparison, we treated each of the ground truth audio-visual captions and LLM-generated audio-visual captions as a prediction and their corresponding ground truth audio and visual captions from the same clip as the references. We calculated n-gram overlaps using METEOR Lavie & Agarwal (2007) and ROUGE_L Lin (2004) metrics, semantic similarity using CIDEr Vedantam et al. (2015), SPICE metric Anderson et al. (2016), and the sentence similarity using Sentence-BERT Reimers & Gurevych (2019) cosine similarity ($SBERT_{sim}$). We report our findings in Table 3.

Table 3: Comparison of ground truth (GT) and LLM-generated (LLM) audio-visual captions with the ground truth audio and visual captions.

| Reference | Predicted | METEOR | ROUGE_L | CIDEr | SPICE | $SBERT_{sim}$ |
|---|---|---|---|---|---|---|
| Visual | Audio-visual (GT) | 0.25 | 0.45 | 0.62 | 0.21 | 0.62 |
| Audio | Audio-visual (GT) | 0.14 | 0.27 | 0.14 | 0.10 | 0.36 |
| Visual | Audio-visual (LLM) | 0.31 | 0.51 | 0.71 | 0.30 | 0.66 |
| Audio | Audio-visual (LLM) | 0.23 | 0.38 | 0.37 | 0.21 | 0.49 |

It can be seen that the LLM-generated audio-visual captions lead to higher similarity to both visual and audio ground truth captions compared to the crowdsourced audio-visual captions. Specifically, the similarity between the LLM-generated captions and the audio ground truth captions is almost 14% higher (in terms of $SBERT_{sim}$) than the similarity between the audio ground truth and the crowdsourced audio-visual captions.

## 3 EXPERIMENTS

In this section, we present various captioning and retrieval tasks realized with the AVCaps dataset. Specifically, we build audio-visual captioning and retrieval models and show how modality-specific captions can be used to evaluate the ability of these models to capture information from both audio and visual content. These experiments can also serve as a baseline for methods that will be developed in the future using this dataset.

### 3.1 CAPTIONING

Captioning is the task of generating a natural language description for an input signal like an image or audio. A typical deep-learning captioning model consists of an encoder and a decoder. The encoder processes the input signal, extracts features, and creates a representation. The decoder then generates the caption in an auto-regressive manner using this encoded representation. For unimodal tasks like image or audio captioning, encoders are selected based on the input modality. ResNet He et al. (2016) or VGG Simonyan & Zisserman (2014) models are common for image feature extraction, while PANNs Kong et al. (2020) or VGGish Hershey et al. (2017) are used for audio. With the advancements in natural language processing, transformers have become the gold standard for decoders to generate textual descriptions.

**Audio-visual captioning:** Using our AVCaps dataset, we developed an audio-visual captioning system. This model consists of two encoders - PANNs Kong et al. (2020) for the audio feature extraction and ResNet-3D Tran et al. (2018) for the visual feature extraction. We used pre-trained PANNs finetuned on the Clotho dataset Drossos et al. (2020) for the audio retrieval task as a part of the DCASE 2023 challenge [3], and for visual feature extraction we used the ResNet-3D model

---
[3]https://dcase.community/challenge2023/task-language-based-audio-retrieval

pre-trained on Kinetics 400 dataset Kay et al. (2017). Our decoder is a 6-layer transformer model trained to predict the next token. We used the GPT-2 tokenizer Radford et al. (2019) which has 50k tokens, to tokenize the input text fed to the decoder.

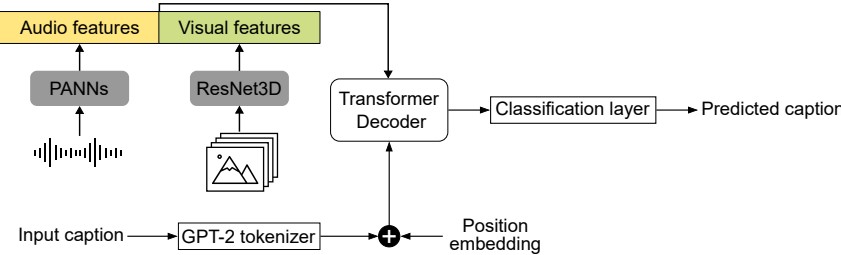

Figure 2: Audio-visual captioning model architecture

Audio files were cropped or padded to two minutes and sampled at 16 kHz. Log-mel spectrograms with 64 mel bins were extracted and fed into the pre-trained PANNs model. The PANNs encoder produces an audio embedding $\mathbb{R}^{75 \times 2048}$, where 75 is the temporal dimension and 2048 is the feature dimension. Similarly, the visual clips were cropped or padded to two minutes, sampled at 5 frames per second, and fed to the ResNet-3D model. The ResNet-3D outputs a visual representation $\mathbb{R}^{75 \times 512}$, where 512 is the temporal dimension and 512 is the feature dimension. The output from the audio encoder is transformed via a learnable linear layer to a $75 \times 512$ representation and then downsampled to match the temporal resolution of the visual embedding. These are then concatenated along the feature dimension and passed to a 6-layer transformer decoder with 1024 attention units. The decoder generates audio-visual captions using the joint audio-visual representation. The audio-visual captioning model architecture is shown in Figure 2. We trained two models, one using the ground truth audio-visual captions and the other using the LLM-generated audio-visual captions. We evaluated these models on the test split using their corresponding audio-visual captions as references. We report the performance of both models in Table 4.

Table 4: Performance of audio-visual captioning models on the AVCaps dataset. **Audio-visual (GT)** and **Audio-visual (LLM)** refer to the models trained and evaluated on ground truth and LLM-generated audio-visual captions respectively.

| Model | METEOR | ROUGE_L | CIDEr | SPICE | SPIDEr |
|---|---|---|---|---|---|
| Audio-visual (GT) | 0.20 | 0.44 | 0.37 | 0.11 | 0.24 |
| Audio-visual (LLM) | 0.18 | 0.36 | 0.33 | 0.14 | 0.23 |

**Evaluation of audio-visual captioning:** While the model trained and evaluated on ground truth captions achieves higher metrics overall, likely due to the shorter average caption length compared to the LLM-generated captions, the latter model shows a stronger ability to capture both audio and visual information. To validate this, we conducted a comparative analysis of the predicted captions from both models against the ground truth audio and visual captions. The results, shown in Table 5, indicate that both models effectively capture visual cues within the clips, but the model trained on LLM-generated captions performs significantly better in identifying audio cues.

Table 5: Comparison of predicted audio-viusal captions (**Predicted**) against the audio and visual ground truth captions (**Reference**).

| Predicted | Reference | METEOR | ROUGE_L | CIDEr | SPICE | SBERT$_{sim}$ |
|---|---|---|---|---|---|---|
| Audio-visual (GT) | Audio | 0.15 | 0.34 | 0.17 | 0.08 | 0.31 |
| | Visual | 0.21 | 0.45 | 0.42 | 0.12 | 0.44 |
| Audio-visual (LLM) | Audio | 0.20 | 0.35 | 0.16 | 0.15 | 0.45 |
| | Visual | 0.19 | 0.36 | 0.21 | 0.17 | 0.43 |

These findings reinforce our observation that when humans process audio-visual content, they tend to focus more on visual information than auditory details. However, when they described the audio

and visual modalities separately, and this information is later integrated (as in the case of LLM-generated captions), the captioning model can learn from both modalities. In this way, by utilizing modality-specific captions, we can assess how well multimodal models capture information from different input modalities and improve their capacity to produce more comprehensive and balanced captions.

**Crossmodal captioning:** A unique feature of our dataset is crossmodal captioning, where models are trained on inputs from one modality and ground truth labels from another. For instance, audio captions can be generated from visual inputs and vice versa. Our dataset allows validation of these generated captions using ground truth from the missing modality, making it valuable for generative AI applications, such as augmenting videos with realistic sounds or generating videos from audio. For our experiments, we trained two crossmodal captioning models: one that generates audio-visual captions from audio inputs, and another that generates audio-visual captions from visual inputs. Both models were trained and evaluated using LLM-generated audio-visual captions. The performance of these models is presented in Table 6. The results show that the model generates audio-visual captions more effectively from visual input than from audio input. This may be because visual modality often contains more information than audio modality. This is reflected in the 21% higher SPIDEr score for the model using visual input, indicating greater semantic similarity with the LLM-generated audio-visual captions.

Table 6: Comparison of crossmodal captioning models trained and evaluated on audio-visual captions (**Reference**) when a single modality is given (**Input**).

| Input | Reference | METEOR | ROUGE_L | CIDEr | SPICE | SPIDEr |
|-------|-----------|--------|---------|-------|-------|--------|
| Audio | Audio-visual (LLM) | 0.16 | 0.36 | 0.24 | 0.11 | 0.18 |
| Visual | Audio-visual (LLM) | 0.17 | 0.37 | 0.30 | 0.13 | 0.21 |

**Evaluation of crossmodal captioning:** To assess how effectively the predicted audio-visual captions capture the information in the missing modality, the predicted audio-visual captions are compared with the ground truth captions of the missing modality and presented in Table 7. The results indicate both the models can partly capture the information from the missing modality. It can again be seen that generating captions related to audio is relatively easier when provided with visual input compared to the reverse scenario. Leveraging advanced pre-trained LLM decoders and techniques such as self-supervised learning and data augmentation, the crossmodal performance can be further improved. In addition to these experiments, we developed unimodal captioning tasks, such as audio and visual captioning. These experiments are presented in Appendix E.

Table 7: Comparison of the predicted audio-visual captions (**Predicted**) when a single modality is given (**Input**), with the missing modality ground truth captions (**Reference**).

| Input | Predicted | Reference | METEOR | ROUGE_L | CIDEr | SPICE | SBERT$_{sim}$ |
|-------|-----------|-----------|--------|---------|-------|-------|---------------|
| Audio | Audio-visual (LLM) | Visual | 0.17 | 0.35 | 0.14 | 0.10 | 0.36 |
| Visual | Audio-visual (LLM) | Audio | 0.22 | 0.36 | 0.17 | 0.13 | 0.44 |

## 3.2 RETRIEVAL

Retrieval is the task of identifying and returning relevant information based on an input query. For example, in text-based image retrieval, images from a database that corresponds to a textual query are returned. Similarly, retrieval tasks can be extended to return audio clips, audio-visual clips, and so on. One common way in deep learning to build retrieval models is to have two encoders. A textual encoder encodes the information in the query text and an audio or visual encoder encodes all the audio or visual files in the database. Then the cosine similarity between the text representation and all the audio or visual representations is calculated and the files with top similarity scores are returned. To train these retrieval models, a contrastive training approach is adopted, wherein the similarity between media files and their corresponding ground truth descriptions is increased while the similarity with the other textual descriptions and other media files in the database is decreased.

**Audio-visual retrieval:** Similarly to captioning models, we developed audio-visual models. The task is to retrieve videos that correspond to audio-visual descriptions. Figure 3 shows the architecture of our audio-visual retrieval model. The audio and visual embeddings from the PANNs and ResNet-3D encoders are averaged over the temporal dimension, concatenated, and mapped to a fixed-size embedding of size 512 using a learnable linear layer. To facilitate the learning of similarity between the representations, InfoNCE van den Oord et al. (2019) loss was applied among these fixed-size embeddings to aid contrastive learning. The contrastive training was performed on mini-batches of size 32.

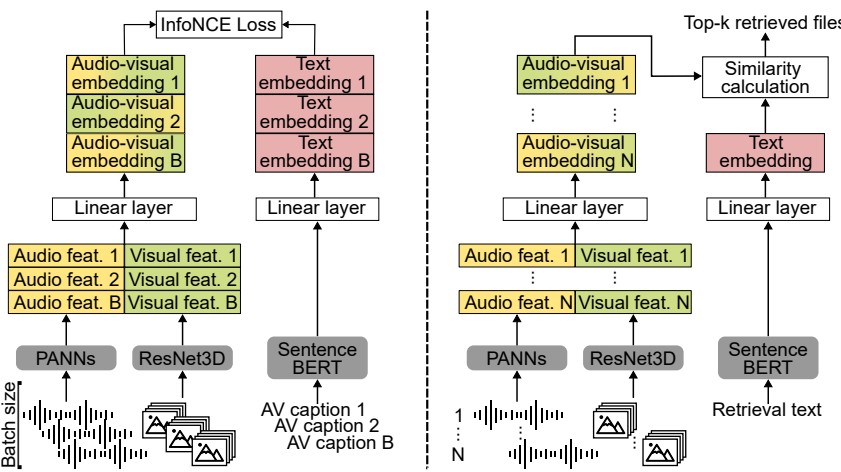

Figure 3: Audio-visual retrieval model. **Left**: Training. **Right**: Inference

The retrieval task is typically evaluated using the recall metric ($recall@k$, where $k$ represents the number of top results taken into account for the evaluation.). Table 8 presents the performance of our audio-visual retrieval model trained with both ground truth and LLM-generated audio-visual captions. The results indicate that LLM-generated captions outperform ground truth captions, as they encapsulate information from both modalities, enhancing the retrieval of video clips with similar visual content.

Table 8: Performance of audio-visual retrieval models on the AVCaps dataset.

| Text Query | Recall@1 | Recall@5 | Recall@10 |
|---|---|---|---|
| Audio-visual (GT) | 0.12 | 0.36 | 0.49 |
| Audio-visual (LLM) | 0.16 | 0.41 | 0.57 |

**Crossmodal retrieval:** A crossmodal retrieval task is again a unique feature of this dataset. It can be used to retrieve visuals based on audio captions and vice-versa. These crossmodal experiments are presented in Appendix E. We performed audio and visual retrieval based on LLM-generated audio-visual captions and compared them with retrieval based on corresponding ground truth captions. The results of these experiments are presented in Table 9. It can be seen that the LLM-generated captions which combined information from the audio and visual captions retrieve audio and visual files better than the models trained with the corresponding ground truth captions.

Table 9: Comparison of unimodal and crossmodal retrieval on the AVCaps dataset

| Modality | Text Query | Recall@1 | Recall@5 | Recall@10 |
|---|---|---|---|---|
| Audio | Audio | 0.05 | 0.23 | 0.36 |
| Visual | Visual | 0.18 | 0.44 | 0.57 |
| Audio | Audio-visual (LLM) | 0.08 | 0.29 | 0.43 |
| Visual | Audio-visual (LLM) | 0.18 | 0.47 | 0.62 |

## 4 POSSIBILITIES WITH THE DATASET

**Representation Learning:** Traditionally, multimodal representation learning has concentrated on two modalities, such as image and text (e.g., CLIP) Radford et al. (2021) and audio and text (e.g., CLAP) Elizalde et al. (2023). Some studies, such as Wu et al. (2022) employ a two-step approach to create a shared representation across audio, visual, and textual modalities. First, a model is trained to align images and text in a shared space using an image-text dataset. The text/image encoder is then frozen or fine-tuned to create a shared representation with audio using a separate audio-text or audio-image dataset. A key disadvantage of this approach is the suboptimal alignment due to the independent training phases and different datasets used in these phases. In contrast, our dataset provides paired audio, visual, and audio-visual captions, enabling the development of a single-stage shared representation encompassing all three modalities. This approach has the potential to significantly enhance performance in tasks such as video understanding, video classification, captioning, and retrieval.

**Multimodal Question Answering:** Using LLMs and modality specific captions, our dataset enables the creation of multimodal question-answering datasets by forming questions and answers about different aspects of the video content, such as sound events, visual actions, or combined audio-visual occurrence. We can then train models that provide contextually accurate responses about various aspects of video clips compared to previous approaches that often focus on isolated modalities.

**Benchmarking and Metrics:** This dataset establishes a new benchmark for evaluating video captioning models. With ground truth available for each modality, we can assess how effectively these models capture and integrate information from both audio and visual inputs, enabling the development of new metrics for video captioning. This capability is essential for advancing the state-of-the-art in multimodal scene understanding.

**Rich Video Descriptions:** Using captions from multiple modalities, incorporating additional metadata from the VidOR dataset such as spatiotemporal bounding boxes, object detection, and sound event detection models, we can generate comprehensive and descriptive video summaries. Unlike existing datasets that focus on a single modality, our dataset combines information from various sources, providing a richer context for a more holistic understanding of the video.

**Generative Modeling:** State-of-the-art video generation models lack the capability to produce videos accompanied by their corresponding audio. However, this dataset facilitates the training of such models since it has ground truth captions for individual modalities. We can train models to generate visual content for a given audio caption, and conversely, generate audio from a visual caption. This capability significantly enhances the potential for multimodal content generation.

## 5 CONCLUSION

In this paper, we presented AVCaps, an audio-visual captioning dataset consisting of 2061 video clips, each paired with up to five human-annotated captions for the audio, visual, and audio-visual content. To address the bias toward visual cues in crowdsourced audio-visual captions, we included three audio-visual captions generated by large language models (LLMs) based on the ground truth annotations. Using multimodal and crossmodal captioning and retrieval experiments, we demonstrated the effectiveness of modality-specific captions for model evaluation. Our captioning experiments showed that models trained on LLM-generated audio-visual captions capture audio information more effectively, achieving 14% higher Sentence-BERT similarity on ground truth audio captions compared to models trained on crowdsourced audio-visual captions. In retrieval experiments, we found that audio-visual text queries improve recall@10 in audio retrieval by 7% and visual retrieval by 5%, demonstrating the advantage of incorporating crossmodal information. Finally, we discussed potential applications of the dataset in advancing multimodal learning, highlighting the value of AVCaps for future research in this domain.

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

# A  EXAMPLES FROM THE DATASET

In this section, we include some examples from the dataset. Figures 4, 5, and 6 show sample video clips present in the dataset. We also show the corresponding audio, visual, and audio-visual captions which were crowdsourced from human annotators. Note that to collect the audio captions, the audio content was extracted from the videos, and annotators only had access to the audio clip and not to the visual content. Similarly, the audio content was removed from the videos and shown to the annotators for the visual captioning task. For the audio-visual captioning task, the annotators had access to both modalities. We also include the LLM-generated audio-visual captions for these corresponding examples.

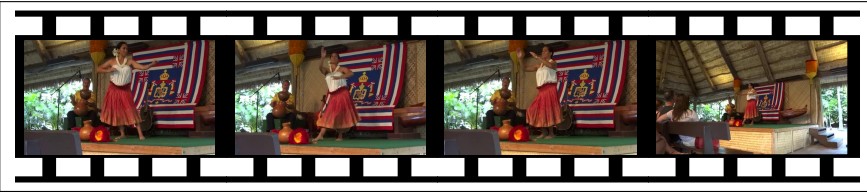

Figure 4: An example video from the dataset.

| Caption type | Captions |
| --- | --- |
| Audio | A person sang a song in front of everyone, and finally, they all clapped. Someone is singing loudly, and people are encouraging and clapping. A man is singing and making music on their own. People are singing, dancing, and playing music. A man is making sounds and singing a song. A siren sound is heard in the background, and people are clapping at the end. |
| Visual | The woman is good at dancing. A lady is performing a dance on the stage. A woman is dancing on the stage wearing traditional dress and accompanied by live music. |
| Audio-visual (GT) | The girl is dancing superbly. A lady is dancing on the stage to music. The girl is dancing and the man is singing with a very nice voice. A woman shows a dancing performance on stage in front of people. |
| Audio-visual (LLM) | A woman is dancing on the stage in traditional dress while a man sings and people clap. A lady performs a dance on stage accompanied by live music and audience applause. A woman dances superbly on stage as a man sings and a siren sounds in the background. |

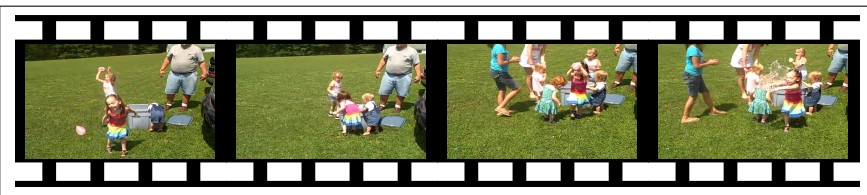

Figure 5: An example video from the dataset.

| Caption type | Captions |
|---|---|
| Audio | A lady is speaking to the children. Other children are also speaking. A child is shouting. A woman is talking to another woman while children are playing and talking loudly in the background. The children are talking loudly, and someone is being noisy and clapping. |
| Visual | These kids are playing on a playground. The kids are happily seen playing with water balloons. A man and a woman are also seen in the video. The babies are playing on the ground. The kids are playing a water ball game. |
| Audio-visual (GT) | This is a children's park where all the children are playing with water balloons and enjoying themselves. The children are playing in the garden. The children are playing in the park. The children are playing with water balloons and they are really enjoying it. |
| Audio-visual (LLM) | Children are playing with water balloons in a park, enjoying themselves while a woman talks to another woman. Kids are happily playing with water balloons in a park, with a woman speaking and children talking loudly in the background. Children are seen playing with water balloons in a park, with a woman conversing and kids making noise. |

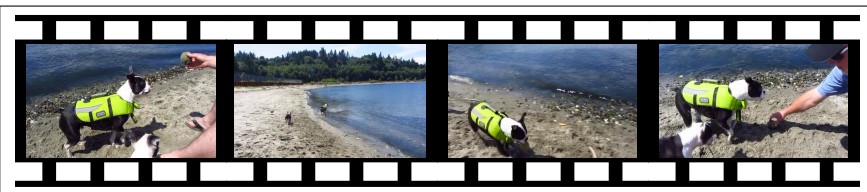

Figure 6: An example video from the dataset.

| Caption type | Captions |
|---|---|
| Audio | The dog wants to play catch with the owner on the beach. A man is talking to a hungry, whining dog while a river flows in the background. A small puppy is whining and a man is taking care of the puppy. A dog was barking in the middle of the river, and a man was helping the dog. |
| Visual | The man is standing on the beach while his dogs are playing. Two dogs are playing on the beach. Dogs are playing with the ball on the beach. The dog is barking in front of the man. The two dogs are playing with the ball. Two dogs are barking near the sea. |
| Audio-visual (GT) | The owner is playing catch with two dogs on the beach. Two dogs are on the beach, and they bark at their owner. He throws a ball into the sea, and the two dogs run to retrieve it. They then give it back to the owner. The dog is playing by the beach. |
| Audio-visual (LLM) | A man is playing catch with two dogs on the beach, throwing a ball into the sea for them to retrieve. Two dogs are barking and playing with a ball on the beach while their owner stands nearby. The owner is interacting with two dogs on the beach, throwing a ball for them to fetch and return. |

## B ANNOTATOR INSTRUCTIONS

In this section, we describe the instructions provided to Amazon Mechanical Turk (AMT) workers for the captioning annotation tasks across different modalities. Figures 7, 8, and 9 illustrate the user interface shown to the workers for the audio, visual, and audio-visual captioning tasks, respectively. The instructions were designed to be straightforward, ensuring that workers could easily grasp the task requirements and generate captions without unnecessary constraints. Additionally, example captions were provided to clarify the expected output for each task.

To reduce noisy annotations, we implemented a strategy where the caption textboxes were activated only after the workers had fully listened to the audio or watched the video. Additionally, we enforced a minimum caption length of five words to ensure more detailed and informative responses from the annotators. It can be noted that for the audio-visual captioning task, we did not have explicit instructions for the annotators to focus on both the audio and visual modalities or to provide captions that include information from both modalities. Hence, the ground truth audio-visual captions emphasized more on the visual aspects compared to the auditory aspects present in the clip.

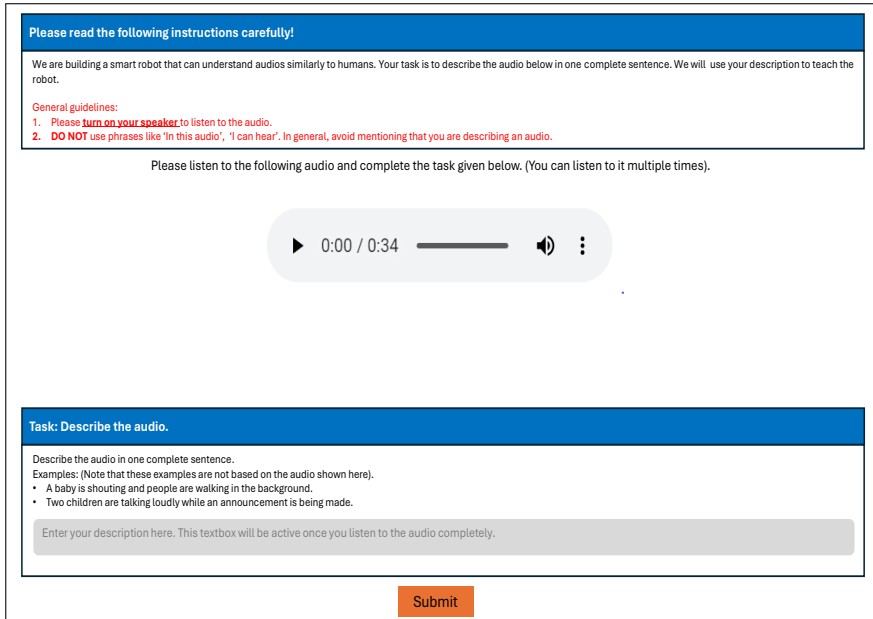

Figure 7: Instructions to AMT workers for the audio captioning task.

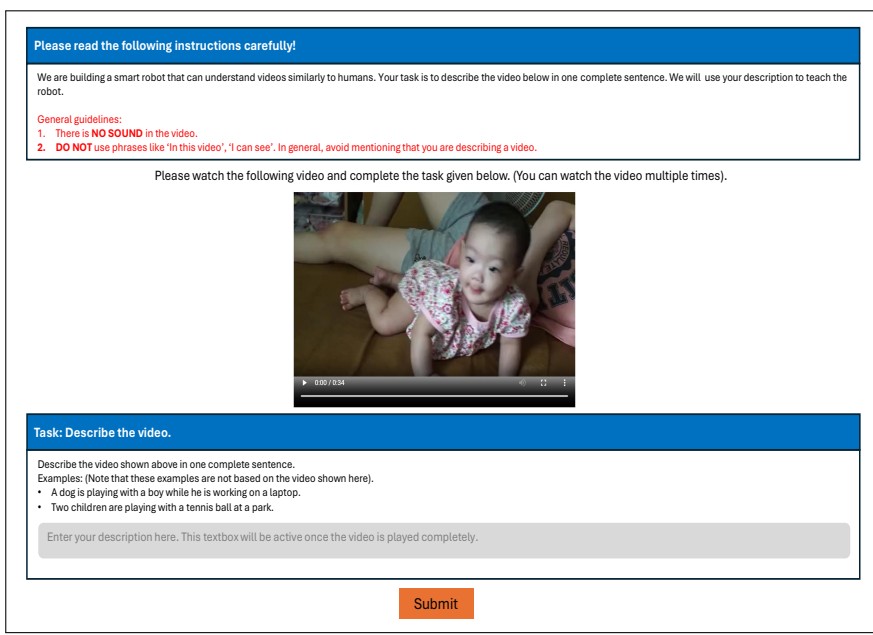

Figure 8: Instructions to AMT workers for the visual captioning task.

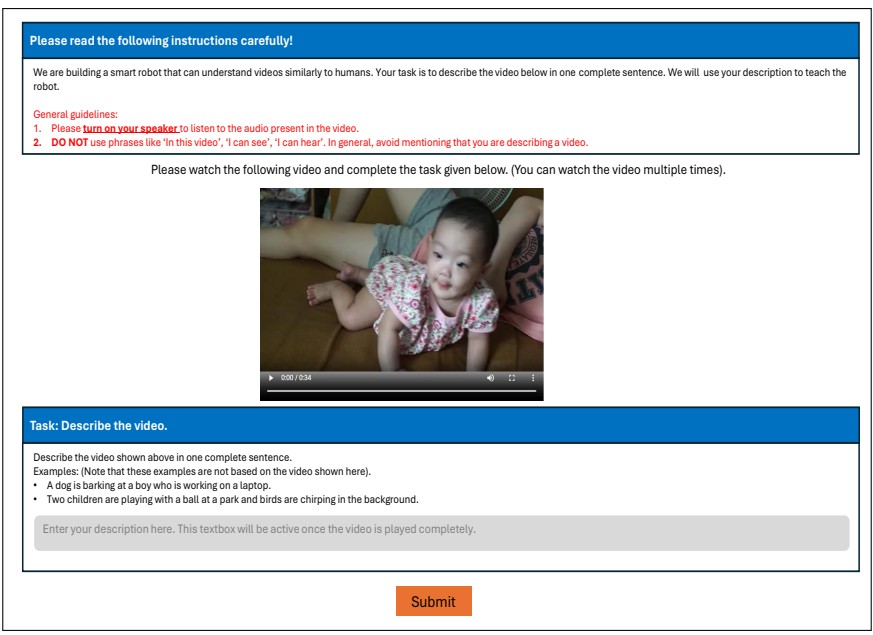

Figure 9: Instructions to AMT workers for the audio-visual captioning task.

## C  AUTOMATED ERROR CORRECTION WITH GPT

As discussed in Section 2.2, we used GPT-3.5 to correct spelling and grammatical errors in the crowdsourced captions automatically. Some examples of the errors in annotated captions and their corresponding GPT-corrected versions are shown in Table 10.

It should be noted that while GPT-3.5 fixed the grammar and spelling in all the captions, in some cases like the example in the last row, it rephrased the caption to give a slightly different context than what was provided by the annotator. To understand the impact of the error-correction process,

Table 10: Crowdsourced captions and their corresponding GPT-corrected versions.

| Crowdsourced Caption | GPT-Corrected Caption |
|---|---|
| Teacher is teaching programming using a projector in front of her class and she is explaining the subject. | A teacher is teaching programming using a projector in front of her class and explaining the subject. |
| Teacher teaching a class and students answreing it. | A teacher is teaching a class and the students are answering. |
| Two adults is talking to a baby in crowded area. | Two adults are engaged in conversation with a baby in a busy location. |

we compared the crowdsourced captions and the error-corrected captions of all the modalities based on their average length in words, readability index, and ease of reading. We present our findings in Table 11. As spelling errors in the crowdsourced captions are fixed, the vocabulary size reduces after the error correction. The average length of the corrected captions increases as the LLM adds missing articles and determinants to these captions. GPT-error-corrected captions use more complex sentence patterns that increase the age and grade levels required to comprehend the captions.

Table 11: Comparison of crowdsourced captions and GPT corrected captions.

| Metrics | Crowdsourced Captions | Corrected captions |
|---|---|---|
| Number of captions | 21456 | 21456 |
| Vocabulary (lemma) | 5555 | 2907 |
| Average length in words | 9.63 | 10.44 |
| Automated readability index | ages: 7 to 9 | ages: 9 to 10 |
| Flesch reading ease | Easy, grade level 6 | Fairly easy, grade 7 |

Finally, we calculated if a sentence is grammatically correct by defining grammar rules. We defined the rules such as there should be noun phrases, verb phrases, prepositional phrases, and a clause. A clause is defined as a group of words that contains a subject and a predicate; we defined it as a noun phrase followed by a verb phrase. We used the Python natural language toolkit [4] to process the sentences. First, we tokenized and tagged the parts of speech in the sentence, then we used the parser to map the sentence into a tree structure and finally identified if the clause is present in the parsed structure. If the label clause is found, the sentence is identified as grammatically correct. Our analysis showed that 38.7% of the crowdsourced captions and 81.3% of the corrected captions were categorized as grammatically correct. It should be noted that the defined grammar rules are too simple, and they are not always able to identify all the grammar structures of a sentence. For example, it can not identify different verbs forms or subordinate clauses.

## D GROUND TRUTH VS LLM-GENERATED AUDIO-VISUAL CAPTIONS

In Section 2.6, we quantitatively showed with various captioning metrics that the ground truth audio-visual captions place greater emphasis on visual cues than on auditory cues. To illustrate this observation, we present an example that highlights this disparity. Additionally, we include the corresponding LLM-generated audio-visual captions, which incorporate information from both modalities.

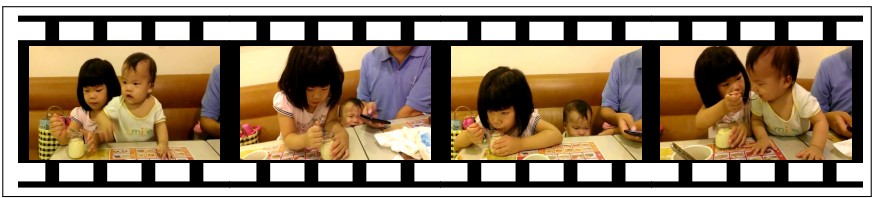

Figure 10: An example video from the dataset.

---

[4] https://www.nltk.org/

| Caption type | Captions |
|---|---|
| Audio | The **baby and** her **mother are speaking** in a cute voice.
The **children are talking to** the **parent**.
A **lady** and a **boy** are **talking**. |
| Visual | Two **children are eating something** on the sofa.
Two **children are drinking beverages**.
There are two children and their father. One child is a boy and the other is a girl. The **girl** is **eating** food while the boy is not eating. The **father** is **using** his mobile **phone** and the **children** are **fighting**. |
| Audio-visual (GT) | Two **children** are **at the table**. The elder one is eating, and the younger one is playing.
A little **girl is eating** food with a spoon and she is **giving food to the baby** next to her.
The **children are fighting at the dining table**.
A **little girl is eating food** and she is **giving** that **food to the baby** near her. |
| Audio-visual (LLM) | Two **children** are **at the table**. The **girl** is **eating food** with a spoon and **giving** some **to** the **baby** next to her while the **father uses** his mobile **phone**.
The **children** are **talking to** their **parent** while the **girl eats** and the **boy plays**; they occasionally **fight**.
A **lady** and a **boy** are **talking** as the **girl eats** and **shares food** with the baby beside her. |

In the above example, objects, attributes, and relationships marked in bold are extracted using the Stanford Scene Graph Schuster et al. (2015). The audio captions capture the conversation between the babies and their parents, including the speech of a woman who is not visible in the visual modality but is audible (sitting on the opposite side of the table). However, this auditory information is not reflected in the ground truth audio-visual captions, which predominantly focus on the visual aspects of the scene, such as a little girl eating and sharing food, while omitting the conversational details. In contrast, the captions generated by Large Language Models (LLMs) incorporate both auditory and visual elements, effectively capturing the conversation and providing a more comprehensive description of the clip.

# E    OTHER EXPERIMENTS

In addition to the experiments presented in Section 3, we developed various other unimodal and crossmodal captioning and retrieval models that are presented in this section.

**Audio captioning:** The audio captioning model takes audio files cropped or padded for 2 minutes, sampled at 16 kHz as inputs, and their corresponding ground truth captions as labels. Log-mel spectrograms with 64 mel bins are extracted and fed into the pre-trained PANNs model, producing an encoded representation $\mathbb{R}^{75 \times 2048}$, where 75 is the temporal dimension and 2048 is the feature dimension. This is transformed via a learnable linear layer to a 75x512 representation and passed to a 6-layer transformer decoder with an attention size of 512 that predicts the next token auto-regressively.

**Visual captioning:** Similarly, a visual captioning model was trained using muted video clips as input and ground truth captions as labels. The clips were cropped or padded to 2 minutes and sampled at 5 fps and visual features were extracted using a pre-trained ResNet-3D. It produces an encoded representation $\mathbb{R}^{75 \times 512}$, where 512 is the temporal dimension and 512 is the feature dimension that is passed to the decoder. The decoder architecture is identical to that used for the audio captioning model. Note that in both cases, the transformer decoders are trained from scratch. The results of the unimodal captioning experiments are presented in Table 12. It can be seen that the audio captioning model performs slightly better in the SPICE metric, while the visual captioning model shows better performance in ROUGE_L and CIDEr.

**Crossmodal captioning:** Next, we developed additional crossmodal captioning models. Specifically, we trained models that generate visual captions using audio input and vice-versa. During the training phase, these models were provided inputs from one modality and captions from the other modality. During inference, the predicted captions are evaluated against the ground truth captions of the cross-modality. In Table 13, we present the performance of these crossmodal captioning models.

Table 12: Performance of unimodal captioning models on the AVCaps dataset.

| Modality | METEOR | ROUGE_L | CIDEr | SPICE | SPIDEr |
|----------|--------|---------|-------|-------|--------|
| Audio    | 0.200  | 0.380   | 0.333 | 0.153 | 0.243  |
| Visual   | 0.195  | 0.424   | 0.416 | 0.117 | 0.267  |

These results suggest that visual modality contains more cues to learn about the audio than audio modality about visuals.

Table 13: Performance of crossmodal captioning models on the AVCaps dataset.

| Input  | Reference | METEOR | ROUGE_L | CIDEr | SPICE | SPIDEr |
|--------|-----------|--------|---------|-------|-------|--------|
| Audio  | Visual    | 0.187  | 0.415   | 0.257 | 0.098 | 0.177  |
| Visual | Audio     | 0.177  | 0.378   | 0.270 | 0.126 | 0.198  |

**Crossmodal retrieval:** Similarly, we performed crossmodal retrieval experiments to retrieve audio and visual files based on visual and audio captions respectively. During training, the similarity between the input signals of one modality and the corresponding captions of the same signal from the other modality is increased using the InfoNCE loss. The results of our crossmodal retrieval experiments are presented in Table 14. It can be seen that visual captions retrieve audio files nearly twice as effectively as audio captions retrieve visual files.

Table 14: Performance of crossmodal retrieval models on the AVCaps dataset.

| Modality | Text Query | Recall@1 | Recall@5 | Recall@10 |
|----------|-----------|----------|----------|-----------|
| Visual   | Audio     | 0.03     | 0.10     | 0.16      |
| Audio    | Visual    | 0.05     | 0.19     | 0.29      |

