# OpenReview forum: "AVCAPS: AN AUDIO-VISUAL DATASET WITH MODALITY-SPECIFIC CAPTIONS"
_ICLR.cc/2025/Conference — ICLR 2025 Conference Withdrawn Submission_

### Official Review · Reviewer_8h56 · 2024-10-31

**Soundness:** 3
**Presentation:** 3
**Contribution:** 2
**Rating:** 3
**Confidence:** 4

**Summary:**

This paper proposes a dataset AVCaps that includes 2061 videos(28.8 h) with audio and visual captions, including around 5 modality-specific captions from multimodal research. They point out the modality gap of the current research dataset, demonstrating that audio-visual captions can help large language models capture audio and visual information more effectively. The effectiveness of these captions is validated through multimodal and crossmodal captioning and retrieval experiments. The author also discusses the potential use of this dataset to advance research

**Strengths:**

1. The dataset includes manually labeled captions by human annotators and LLMs, ensuring high-quality and contextually accurate descriptions.
2. The data cleaning process is thorough, involving both automated error correction and manual relevance screening. This ensures the dataset’s reliability and accuracy.
3. The experiments are meticulously designed, providing audio-visual captioning and retrieval models for evaluating the effectiveness of the proposed dataset, showcasing the practical application and potential of the dataset in advancing multimodal research.

**Weaknesses:**

1. The task definition is somewhat comprehensive but unclear, making it difficult to understand the specific applications of this dataset. What is the purpose of this dataset?
2. The paper highlights that audio-visual captions can improve caption quality, a point already established by several previous works. This redundancy diminishes the novelty of the contribution.
3. The paper claims that "Existing datasets focus on a single modality or do not provide modality-specific captions" while there are many related works that have noticed the importance of multimodality captions, such as Panda, InternVid, and MMTrail, which also eliminate the novelty of this work.
4. The dataset comprises 2061 clips with a total duration of 28.8 hours. This relatively small size raises concerns about its adequacy for supporting the proposed tasks of generation and understanding.

5. Weak Experimental Validation:

a) Audio-Visual Captioning: The proposed model is evaluated only on its dataset. Including comparisons with other datasets or models would provide a more robust validation.

b) The purpose and implications of the single modality evaluation are confusing. Clarifying what this experiment aims to demonstrate would help in understanding its significance.

c) The proposed experiments do not sufficiently support the method’s claims. Additionally, the lack of comparison with open-source models raises concerns about the overall effectiveness of this work.

**Questions:**

1. Could you explain the potential use of this dataset from the aspect of cutting-edge research?
2. What is the difference between AVCaps? Comparison of AVCaps with other audio-visual datasets.
3. Explain what can this scale of the dataset do?
4. More comparison of modern models in your datasets. What is the performance of another model in your dataset and your setting?
5. What is the performance of your models in other datasets?

---

### Official Review · Reviewer_NSgo · 2024-11-03

**Soundness:** 2
**Presentation:** 2
**Contribution:** 1
**Rating:** 3
**Confidence:** 5

**Summary:**

This paper propose a new audiovisual dataset named AVCAPS. Arthor train captioning and retrieval models on proposed datasets and conduct some experiments.

**Strengths:**

1. The dataset is labeled by person, which to some extent ensure high-quality.

**Weaknesses:**

1. The audiovisual caption dataset is not novel. VALOR and VAST datasets have already proposed to label audiable videos with audiovisual captions, so the novelty of this paper is limited.
2. The scale is limited (only 2000+ videos).
3. Trained models use pann and resnet3d as audio or video encoders which is too out-dated.
4. There are not any comparison between proposed model and dataset  with other open-sourced datasets or models in this paper.

**Questions:**

See weakness

---

### Official Review · Reviewer_GDop · 2024-11-03

**Soundness:** 2
**Presentation:** 3
**Contribution:** 2
**Rating:** 5
**Confidence:** 4

**Summary:**

* The work proposed a new audio-visual captioning dataset (AVCaps) with separate textual captions for audio, visual, and combined audio-visual content of video clips.
* AVCaps addresses the limitations of existing datasets, which either focus on a single modality or lack modality-specific captions.
* Multimodal and crossmodal captioning and retrieval experiments demonstrate the value of modality-specific captions in assessing model performance.
* Models trained on LLM-generated audio-visual captions showed a 14% improvement in capturing audio information (via Sentence-BERT similarity) compared to models trained on crowdsourced audio-visual captions.

**Strengths:**

* The proposed AVCaps provide separate captions for audio, visual, and combined audio-visual content, enabling more studies of each modality’s contribution to comprehension, which fills a significant gap in multimodal research.

* The dataset construction seems legit. Also, the dataset reduces common biases where visual content often dominates, ensuring a fair representation of both modalities.

* The dataset supports various future applications, such as multimodal representation learning and GenAI-related tasks. Also, the author claimed it will be available online.

**Weaknesses:**

1. The main concern of the paper is the lack of discussion on related work. As a dataset paper, it should have a related work section to discuss previous progress and the differences from previous works.

   (a). [1] already had the idea of including both visual and audio information in the captioning dataset.

   (b). [2,3,4] also worked on audio video captioning with related models. The work should be discussed and cited.

2. In the experiment, the author builds a simple baseline with ResNet3D and GPT2.

   (a). This leaves the question to the reviewers: why not a more advanced visual encoder, e.g., visual transformer or current VLM, for captioning for components?

   (b). It will be better to show captioning results with a more advanced video captioning baseline, such as LLava, ... on the proposed dataset.

3. One concern for the dataset is the limited size, containing only 2,061 video clips; the dataset may be relatively small for the multimodal representation learning task, as the author mentioned.

4. The evaluation focuses on Sentence-BERT Similarity, which may not fully capture the richness or context of audio-visual information (atomic actions, attributes of objects, ...) and could limit the scope of the evaluation.


[1] Multi-modal Dense Video Captioning

[2] Audio-Visual Interpretable and Controllable Video Captioning

[3] Integrating Both Visual and Audio Cues for Enhanced Video Caption

[4] A Better Use of Audio-Visual Cues: Dense Video Captioning with Bi-modal Transformer

**Questions:**

1. Please address the weakness accordingly.

2. During the dataset construction, the work claimed to have modality-specific captions. However, some audio and visual elements may inherently overlap, possibly challenging models to distinguish the contributions of each modality independently. How does the author determine if the caption is leaning to video or audio?

---

### Official Review · Reviewer_grv3 · 2024-11-04

**Soundness:** 3
**Presentation:** 3
**Contribution:** 2
**Rating:** 5
**Confidence:** 4

**Summary:**

1.The authors selected 2,176 videos from the VidOR dataset and performed some cleaning on the dataset.
2.Manual annotations were made separately for visual, audio, and audio-visual modalities.
3.LLMs were used to generate another set of audio-visual caption annotations based on the manual captions.
4.Several models was designed based on the proposed dataset, and its effectiveness was validated.

**Strengths:**

1.A new caption dataset was created using manual annotations.

2.Captions were created for each modality, and the effects of modality-specific captions were demonstrated.

**Weaknesses:**

1.The selection of evaluation metrics lacks a thorough justification. The authors treated each of the ground truth audio-visual captions and LLM-generated audio-visual captions as predictions, with their corresponding ground truth audio and visual captions from the same clip serving as references. However, there is a lack of additional comparative experiments to demonstrate that a higher similarity between audio-visual captions and audio/visual captions indicates better quality, as it may introduce issues such as information errors and redundancy.

**Questions:**

1. Questions about the rationality of the evaluation metrics used in the experiments.

The authors treated each of the ground truth audio-visual captions and LLM-generated audio-visual captions as predictions, with their corresponding ground truth audio and visual captions from the same clip serving as references. However, evaluating audio-visual captions solely based on their similarity to audio and visual captions presents certain issues.

(1) If similarity to the audio and visual captions were used as evaluation metrics, have they attempted generating LLM audio-visual captions by providing only the audio and visual captions, without the audio-visual caption? What changes in the metrics were observed from this experiment?

(2) What would be the results if a text model, such as GPT-4, were used to generate audio-visual captions by providing it with audio captions and visual captions, and then comparing these generated captions to the reference captions?

2. Questions about the LLM-generated audio-visual captions.

The analysis in Table 1 of the paper is quite interesting. Is there a similar table that analyzes the characteristics of the LLM-generated audio-visual captions, comparable to Table 1? Such an analysis might further highlight the differences between LLM-generated captions and crowd-sourced captions.

---

### Note · Authors · 2024-11-15

I have read and agree with the venue's withdrawal policy on behalf of myself and my co-authors.